# Analysis of the Efficiency of Various Receipting Multiple Access Methods with Acknowledgement in IoT Networks

**Vyacheslav V. Borodin, Valentin E. Kolesnichenko * and Vyacheslav A. Shevtsov**

Moscow Aviation Institute, Volokolamskoe Highway 4, 125993 Moscow, Russia; borodimai@mail.ru (V.V.B.); vashevtsovmai@mail.ru (V.A.S.)
*   Correspondence: vekolesnichenko@mail.ru

**Abstract:** An Internet of things (IoT) network is a distributed set of "smart" sensors, interconnected via a radio channel. The basic method of accessing the radio channels for these networks is Carrier Sense Multiple Access/Collision Avoidance (CSMA/CA), in which access is carried out on the basis of contention, and confirmation of the correct reception of the packet is achieved using a receipt. If the sizes of information packets are small and comparable to the sizes of receipts, then the transmission of receipts requires a significant bandwidth of the channel, which reduces the efficiency of the network. This problem exists not only for IoT networks but also for monitoring systems, operational management of fast processes, telemetry, short messaging and many other applications. Therefore, an urgent task is to develop effective methods of multiple random access in the transmission of short information packets, the size of which is comparable to the size of receipts. To solve this problem, the authors proposed modifications of CSMA/CA random access in which, when packet collisions are detected, a diagnostic message (DM) is generated and transmitted in the broadcast mode. Based on simulation modeling, it is shown that in a wide range of network loads, the proposed random access options provide an increase in network capacity (the number of connected subscribers) of 1.5–2 times compared to the basic CSMA/CA access method when the size of the information packet is an order of magnitude larger than the size of receipts. The variant of access without acknowledgment is also considered, in which, as shown by the simulation results, at sufficiently large loads, the network can go into an unstable state.

**Keywords:** Internet of Things (IoT); multiple access; collision avoidance; receipt; queuing system; LPWAN

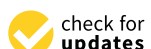



## 1. Introduction

As a result of the development of the production of cheap chips, IoT has become a trend in the information technology market [1–3]. IoT is a distributed network, to which many physical objects are connected through communication and information infrastructure, making them capable of interacting with each other, including other machines or computers [4,5]. The main difference between IoT and other networks is the presence of many physical things and devices other than computing and data-processing devices. The works [3,6,7] reviewed IoT protocols that solve various tasks to meet the necessary requirements for transmission speed, range, frequency range, etc.

This article discusses one of the varieties of IoT: networks with low data transfer rates (up to 250 kbps) with a range of network devices up to 75 m, the organization of which is regulated by IEEE 802.15.4 standard [8]. This standard is focused on the development of cheap network wireless sensors and, in addition to a low data transfer rate, involves the use of relatively short data packets (up to 104 bytes), which is its distinguishing feature.

The basic method of accessing a radio channel in such networks is CSMA/CA [9,10], in which access is based on contention, and confirmation of correct packet reception is achieved using a receipt. With regard to wireless IoT networks, a detailed description of

the CSMA/CA protocol is given in [11,12], so these issues are not considered in this article. A lot of publications in the scientific and technical literature are devoted to the study of the effectiveness and development of various variants of CSMA/CA.

One of the first works on this topic was [13], which proposed using busy-tone multiple-access to prevent collisions in order to reduce packet delay time and increase channel throughput. In [14], a similar Carrier Sense Multiple Access with Collision Avoidance and Detection (CSMA/CAD) method is proposed and analyzed, which can largely solve the problem of packet collision. A similar Carrier Sense Multiple Access with Collision Resolution (CSMA/CR) access method proposed in [15] is based on the fact that the device that detects a collision transmits an interference signal to stop the transmission of other devices, and it transmits immediately, without delay, the data to the radio channel. Such an approach, as it is shown by the calculations performed by the authors, makes it possible to significantly increase the channel capacity in comparison with the conventional CSMA/CA access method. In [16], the Carrier Sense Multiple Access with Enhanced Collision Avoidance (CSMA/ECA) algorithm was explored, which combines the efficiency of channel reservation protocols with the simplicity of random access mechanisms. The article [17] proposes modifying CSMA/CA by using the Ready to send/Clear to send (RTS/CTS) procedure, as a result of which collisions can only occur between signaling messages and not between information packets.

These and other CSMA/CA options allow efficient use of the available radio channel resource, provided that the size of the transmitted packet significantly (by an order of magnitude or more) exceeds the size of the receipt, that is, the receipt transmission time is several tens of times less than the packet transmission time. However, as it was noted above, in networks based on IEEE 802.15.4 standard, the sizes of data packets and receipts are comparable.

Therefore, for IoT networks based on IEEE 802.15.4 standard, an urgent task is to develop effective methods for multiple random access when transmitting short information packets, the size of which is comparable to the size of receipts. This task is relevant not only for IoT networks but also for monitoring systems, operational management of fast processes, telemetry, short messaging and many other applications.

The purpose of this article, devoted to solving this problem, is to develop options for multiple random access when transmitting short information packets that are insensitive to the size of receipts. A distinctive feature of these options is the formation of DM when a collision is detected and its transmission in the broadcast mode. In accordance with this, a simulation model of IoT network was developed, on the basis of which a comparative analysis of the characteristics of the basic CSMA/CA access method (when the size of receipts is an order of magnitude smaller than the size of the information packet) with the random access options proposed by the authors was carried out. As will be shown in Section 3, at a wide range of network loads, such options provide an increase in network capacity of 1.5–2 times compared to the basic method and weakly depend on the ratio of the sizes of receipts and information packets. It is also shown that the lack of acknowledgment can lead to unstable network operation and, as a result, large losses of information packets.

## 2. Theoretical Basis

Figure 1, in accordance with [18], lists the elements involved in IoT, with the left side of the figure showing the various ways of communicating with physical devices (it is assumed that one or more networks support communication between devices). Sensors realize the connection of the physical and digital worlds, providing the collection and processing of information in real time. Gateways allow integrating a network of sensors into a single information network based on radio channels (cellular communications, Wi-Fi, etc.).

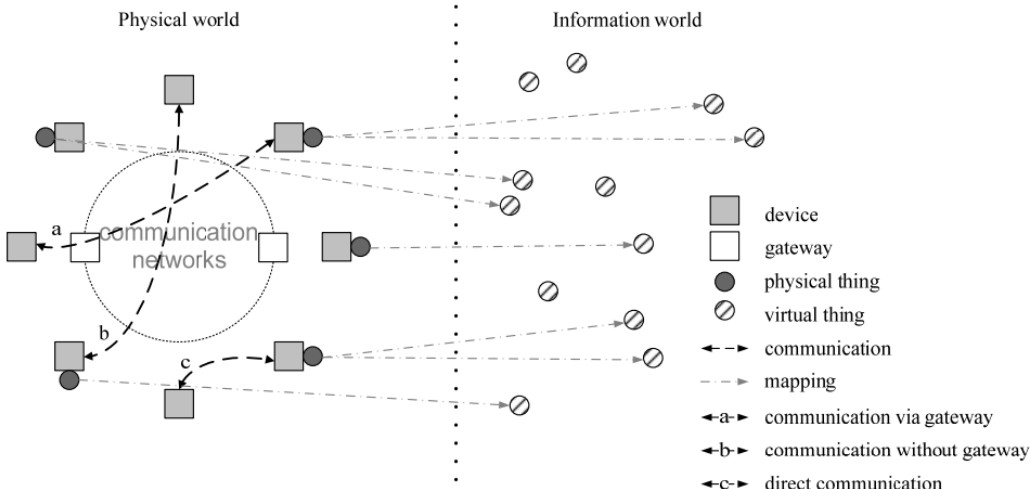

**Figure 1.** IoT elements.

According to regulatory documents [18–20], IoT networks generally contain four basic functional layers: application layer, service and application support layer, network layer and device layer.

For the purpose of this article, two lower levels are interesting:

- Sensors that collect information from physical devices and transmit it in the form of packets to the gateway;
- The gateway that provides transmission of packets from multiple sensors over a common radio channel.

As it was noted above, in accordance with IEEE 802.15.4 standard, access to the radio channel is based on CSMA/CA. Unlike 802.11 standard, the CSMA/CA algorithm does not have a mechanism for pre-requesting the access channel, in which the sending sensor first makes a request for transmission permission before sending an information frame.

When using CSMA/CA, each sensor–sender listens to the channel before sending its packet and, if it is busy, postpones transmission until the channel is free. Once released, the sensor starts transmitting a packet after some random time, which reduces the chance of collisions between packets waiting to be transmitted. However, in this case, collisions are still possible when several sensors access the radio channel simultaneously, which leads to the loss of transmitted packets and/or their delay.

The CSMA/CA algorithm is used in networks without synchronization and in networks with synchronization of access. In completely asynchronous networks, the use of this algorithm is not very effective, while the availability of access synchronization makes it possible to save power consumption of network devices. In addition, in networks with synchronized access, it is possible to use a combination of CSMA/CA with access on a non-contention basis. Therefore, in the future, we will consider only the variant of synchronous access (or slotted CSMA/CA [21]), in which the gateway acts as a coordinator and periodically transmits beacon signals to the channel. Such beacons signal to all devices about the possibility of accessing the channel, while the beacon frame (the time interval between two adjacent beacons) is divided into two parts:

- Active, in which the access procedure is based on CSMA/CA;
- Passive, designed to transmit information packets.

The correctness of the data packet reception is confirmed by an acknowledgment frame based on CRC code, and if an error is detected, the packet is retransmitted. It is also possible to access without acknowledgement, which reduces the power consumption of network devices.

If we denote by $L_p$ and $L_r$ the size of the packet and the receipt, respectively, then the share $\delta$ of the channel bandwidth that is used to transmit the receipt will be equal to (1):

$$\delta = \frac{L_r}{L_r + L_p}. \tag{1}$$

The value of $\delta$ determines the amount of reduction in the channel capacity due to the transmission of the receipt. Given the fact that, in IoT networks, the information flow from sensors often contains one or more samples (the size of the information packet is small and comparable to the size of the receipt), the channel throughput is about half of the maximum value. This circumstance can be tolerated if high-speed communication channels are used and the input traffic is small; however, for low-speed channels, which are typical for IoT networks, the transmission of receipts can lead to network congestion.

Listening to the carrier makes it possible to almost completely eliminate conflicts when accessing the channel, provided that the signal propagation time is negligible compared to the information packet transmission time. In this case, the queuing system M|D|1 can be chosen as a mathematical model of the network and, therefore, the influence of retransmissions and distortion of information packets with such small signal propagation times on the network efficiency can be ignored.

However, if the propagation time of the radio signal cannot be neglected (which is typical for LPWAN networks [22,23]), then collisions of information packets occur, which are most often determined by acknowledging each successfully transmitted packet. The absence of a receipt indicates to the transmitting node that the packet it transmitted was corrupted and that it needs to be retransmitted.

Figure 2 shows the dependence of the average packet transmission waiting time $W$ in CSMA/CA mode with the number of sensors $N = 80$, where $S$ is the amount of primary traffic (the average number of packets generated by nodes during one frame), and $t_p$ is the signal propagation time normalized relative to the frame duration. The simulation results were obtained under the condition that the volume of the information package is 10 or more times bigger than the volume of the receipt. We will call such an access method the basic one.

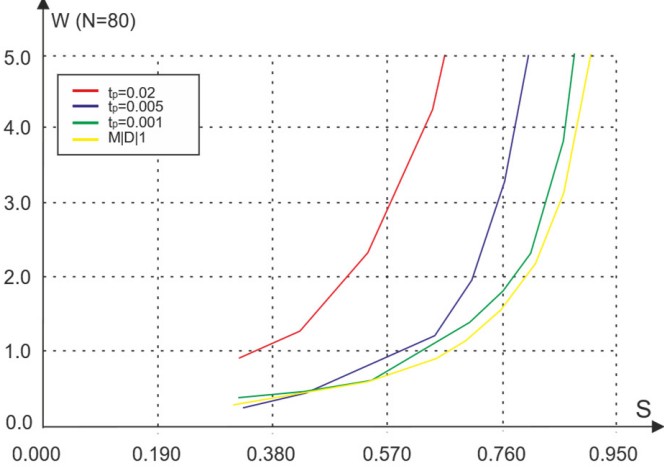

**Figure 2.** Dependence of the average waiting time for different values of the signal propagation time.

The same graph shows a similar dependence obtained for the queuing system M|D|1. These dependencies are obtained using a network simulation model, a brief description of which is given in the next section of the article.

## 3. Materials and Methods

To study the probabilistic and temporal characteristics of multiple random access during the transmission of short information packets, a simulation model [24] was used,

which was originally developed for network research and modernized by the authors in relation to CSMA/CA. The need to use such a model is due to two main factors:

- Due to the complexity of the network and many random factors, the use of known analytical models is not possible;
- Existing programs for network modeling (GPSS [25], OPNET [26], NS2 [27], etc.) do not fully take into account such network features as the presence of various types of interference, service channels, redundancy, routing methods and acknowledgement.

*3.1. Model Description*

The model is designed to study an adaptive packet-switched radio network. When modeling, it is assumed that *N* network nodes are located inside the area (location area) of a given shape (square, cube, etc.) of a unit area (volume). It is possible to form several types of structures (random, hierarchical, linear, ring, etc.)

The model provides for the transmission of packets using random access, as well as cyclic access with time or frequency division of channels. The probability of packet collision is determined taking into account the impact of noise, interference caused by the simultaneous transmission of packets by several nodes, the energy of the equipment and the coordinates of the nodes. To recognize collisions, acknowledgment procedures are modeled, including the possibility of distortion or collisions of the receipts. Each network node can be a source of messages. The model makes it possible to form general recurrent flows, including Poisson ones.

The following adaptive control mechanisms are implemented in the model:

- Management of the period of retransmission of distorted packets depending on the network traffic and the energy potential of the communication channel;
- Access control depending on the traffic and the number of network nodes;
- management of the operation of the service communication channels and its interaction with the information channel.

The model implements processes that provide, under certain conditions, an increase in the efficiency of the network, in particular, grouping packets into messages and channel reservation, as well as changing the ratio between the transmission rates of the main and service channels.

*3.2. Model Composition*

Functionally, the model contains the following blocks:

- Block for setting initial data for modeling;
- Modeling block;
- Blocks for processing and displaying simulation results.

As follows from the above description of the composition of the model, at the first stage of modeling, the selection of initial data at various levels of the network (physical, channel, network, etc.) is carried out. At the second stage, simulation modeling of the system behavior is carried out in order to determine its probabilistic and temporal characteristics (time delays, packet loss probability, number of retransmissions, etc.). At the third stage, the simulation results are processed and presented in the form of graphs, tables and statistical data.

3.2.1. Initial Data Setting Block

The input parameters of the block for setting the initial data are determined for four levels of the open systems interaction model.

At the physical level, the following main parameters are defined:

- Number of network nodes;
- Energy potentials of the radio link;
- Number of transmission channels, methods of sealing and fixing channels;
- Models for calculating the probability of distortion of symbols and packets in general.

The following parameters are defined at the link layer:

- Access methods;
- Packet acknowledgment methods;
- Operation algorithms and service channel parameters.

The network layer defines the following:

- Composition of metrics for packet routing;
- Parameters of network structure changes;
- Routing methods.

The transport layer defines the following:

- Structure of the transmitted messages and the method of splitting the message into packets;
- Redundancy options for channels for data exchange between nodes.

At the level of application processes, the following are set:

- Characteristics of the message source (intensity of message flow, distribution of message size, distribution of time between adjacent packets in a message);
- For each source of messages, the recipient node of the corresponding message is indicated.

### 3.2.2. Simulation Block

Simulation modeling is performed by events on a given time interval of the network operation. It is possible to study stationary and non-stationary processes of network behavior with a given allowable statistical error.

The model allows determining the statistical characteristics of the following quantities:

- Packet transmission waiting time in the node;
- Packet transmission time over the network from the source to the consumer;
- Packet retransmission time and number of retransmissions;
- Probability of packet loss, etc.

The model includes mechanisms for the following:

- Recognition of the network transition to unstable state;
- Reduction in modeling time and obtaining express results;
- Optimization of network parameters;
- Display during the simulation of the network parameters.

### 3.2.3. Blocks for Processing and Displaying Results

Blocks for processing and displaying modelling results form given statistics based on the obtained results (distribution laws, average values, dispersions, etc.) and display the obtained results in the form of graphs and tables.

## 4. Results

The characteristics of various obtained access methods using the simulation model will be compared with the characteristics of the basic access method (Figure 2). To evaluate the effectiveness of access methods, we will use the following set of indicators:

$W$ is the average packet transmission waiting time (the average difference between the moment of successful packet transmission over the channel and the moment of packet arrival at the node);

$S_{max}$ is the capacity of the network (the maximum input traffic, at which the value of the average packet transmission waiting time is not worse than the specified one);

$K_v$ is the coefficient of variation in waiting time (the value equal to the ratio of the standard deviation of the waiting time to the average value) (2):

$$K_v = \frac{\sigma}{W}, \tag{2}$$

$P_{ls}$ is packet loss probability (the proportion of lost packets in relation to all transmitted packets).

### 4.1. Analysis of Characteristics of the Access Method without Packet Acknowledgment

The access algorithm, in which there is no acknowledgment, makes it possible to practically eliminate the decrease in network capacity. Each node listens on the channel before transmitting the packet and, after its release, transmits the packet after a random time. A packet collision is possible when two or more nodes start transmitting their packets at the same time. Mangled packets are removed from the network (lost). In addition to packet loss, with this access method, there will also be a packet transmission delay caused by waiting for the channel to become free.

The input parameters of the model are the primary traffic $S$ (the average number of packets generated by nodes during one frame), the number $N$ of nodes (sensors) of the network and the signal propagation time $t_p$. Also, for simplicity, we will assume that all sensors are the same and form a Poisson packet stream.

We first consider the influence of the signal propagation time $t_p$ on the characteristics of the network without acknowledgement. Figures 3 and 4 show the dependence of the probability of loss and the average waiting time of packet transmission on traffic $S$ and the signal propagation time $t_p$ in the absence of acknowledgement.

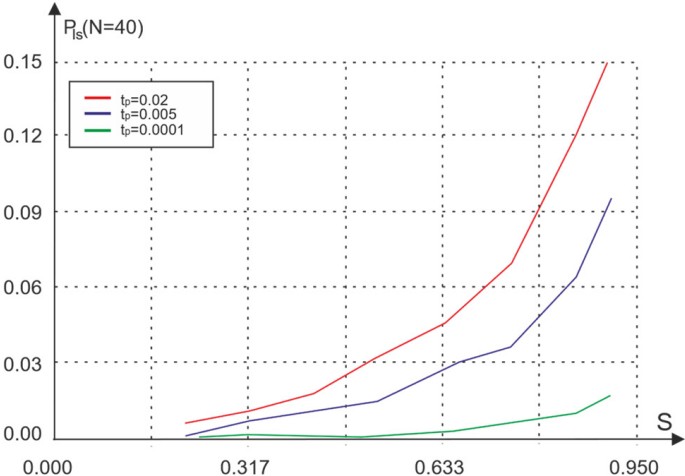

**Figure 3.** Dependence of the loss probability $P_{ls}$ on traffic $S$ depending on the signal propagation time $t_p$.

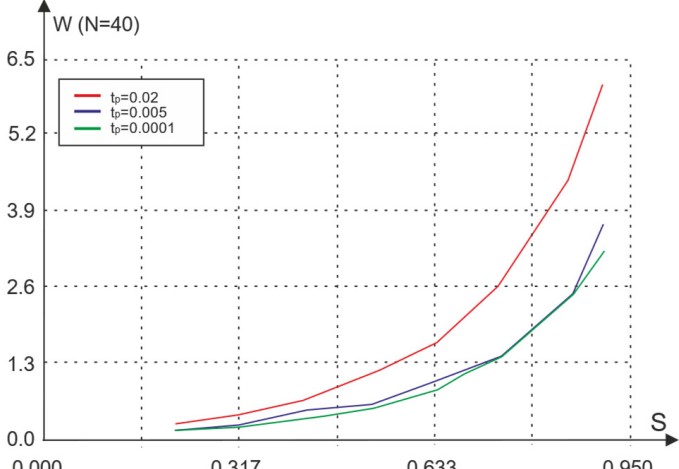

**Figure 4.** Dependence of the average transmission time $W$ on the traffic $S$ depending on the signal propagation time $t_p$.

Comparison of the presented results with the simulation results for the basic access method (Figure 2) shows that the lack of acknowledgment increases the network capacity but leads to packet loss.

Below, we will consider random access methods that either significantly reduce or completely eliminate packet loss. These access methods generate additional service traffic only when packet collisions are detected. Since the probability of collision is small, it is expected that the additional traffic caused by the transmission of such control information will not lead to a significant decrease in network efficiency.

### 4.2. Analysis of the Characteristics of the Access Method with Unrestricted Acknowledgment Collisions

One node is assigned in the network that monitors the traffic between all sensors and determines the fact and time of a collision. In particular, such a dedicated node can be a gateway that ensures the interaction of IoT network sensors with external networks and end users. In the event of a collision, the gateway generates DM containing information about the time when the collision occurred. In order to prevent losses, DM also contains information about the transmission time of DM. The message is transmitted by the gateway according to the general scheme of random access with listening to the carrier. If the transmission of DM occurred without distortion, it is accepted by all nodes.

Each node, in turn, temporarily stores all transmitted packets. If the node receives DM, it then selects from the list of previously transmitted packets those of them whose transmission time coincides with the distortion moments. These packets are passed to the node's output queue for retransmission. The rest of the packets with a later transmission time remain in the queue for retransmission.

The advantage of the proposed access scheme is that it does not allow packet loss. However, we note that DM in the general case can contain information about a large number of distortions, which leads to an increase in the size of DM. In this regard, when implementing this access scheme, it is necessary to provide mechanisms for reducing the amount of transmitted information in order to limit the size of DM. We will consider an idealized case and assume that the size of DM is comparable to the size of an information packet for any number of packet distortions.

The simulation results of the proposed access algorithm are shown in Figures 5–8. As the simulation results show (Figure 5), the average waiting time is almost unchanged compared to the basic access method. At the same time, the number of nodes has practically no effect on the efficiency of functioning.

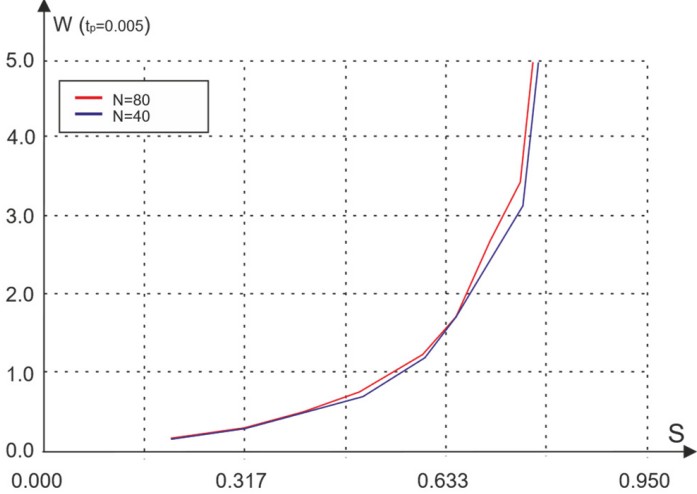

**Figure 5.** Dependence of the average waiting time on traffic for different numbers of nodes.

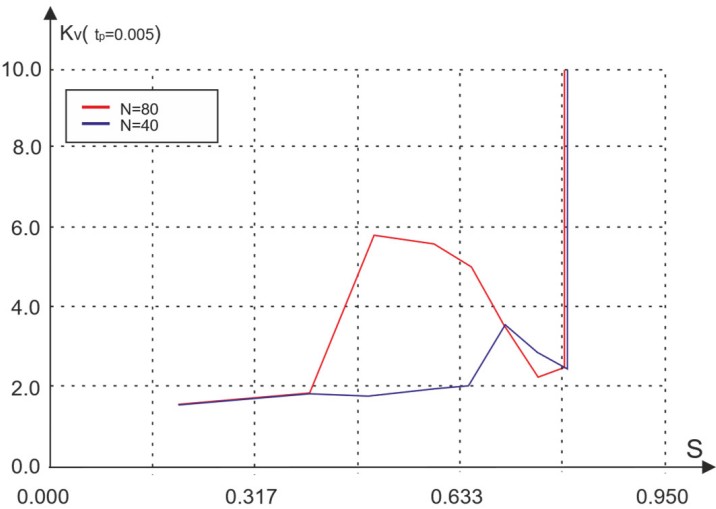

**Figure 6.** Dependence of the variation coefficient of waiting time on traffic for different numbers of nodes.

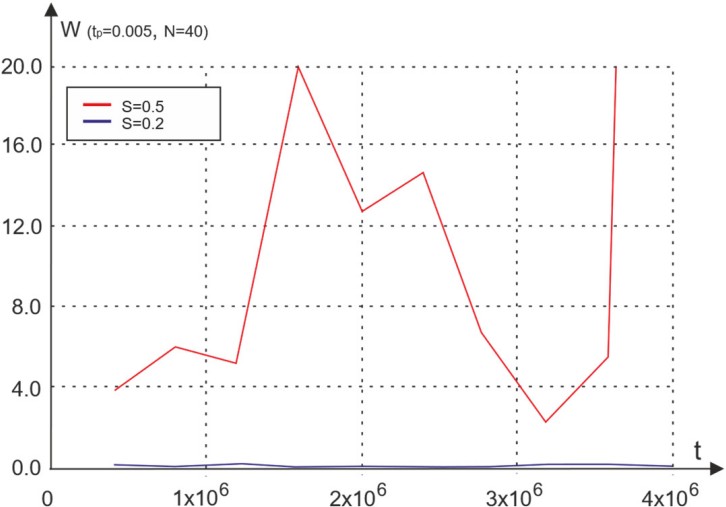

**Figure 7.** Dependence of the waiting time on the normalized operation time for various traffic values.

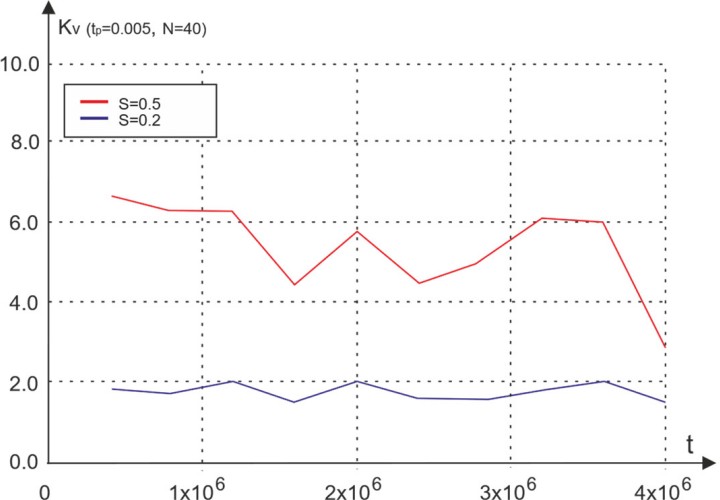

**Figure 8.** Dependence of the coefficient of variation in the waiting time on the operating time for two levels of traffic.

However, a feature of the considered access method is that, due to the distortion of DM, relatively rare but long delays in the retransmission of packets are possible. This circumstance is reflected in the dependence of the coefficient of variation in the waiting time on the amount of input traffic, presented in Figure 6.

The above graphs illustrate the non-monotonic dependence of the coefficient of variation on the amount of traffic, and for a relatively small number of nodes ($N = 40$), this property is more pronounced than in the case of a large number of nodes ($N = 80$). Such a change in the coefficient of variation indicates an unstable mode of network operation, turning into a blocking mode [28].

This leads to a periodic spontaneous significant deterioration in performance indicators with subsequent restoration of efficiency to normal values.

As an example, Figure 7 shows the graphs of the average packet transmission waiting time versus the operation time $t$, normalized with respect to $t_p$, for two different traffic values, $S = 0.2$ and $S = 0.5$. If, at low traffic ($S = 0.2$), the network behavior can be called stationary and the efficiency does not change on average with time, then with traffic growth, the network behavior is clearly undulating, becoming non-stationary and unstable.

The non-stationary unstable behavior of the network, as follows from Figure 8, corresponds to values of the coefficient of variation that are significantly greater than one.

Thus, the use of access with unlimited collision acknowledgment eliminates packet loss and the dependence of the network capacity on the size of the packet or receipt. However, with this access method, the network can enter an unstable state with medium and high traffic, which reduces the overall efficiency of the network.

### 4.3. Analysis of the Characteristics of the Last Collision Acknowledgment Access Method

It is possible to eliminate the above-mentioned disadvantages by limiting the time spent by corrupted packets in the node. In this case, only the packets that received DM immediately after the collision are retransmitted. If DM is corrupted, then all packets that were corrupted are lost. In fact, this means that DM contains only the time of the last collision of information packets. It is possible to implement a method in which each packet, after transmission, waits for DM to arrive for some limited time. If DM is received in this interval, then the packet is retransmitted. Otherwise, it is removed from the node, and the packet is lost.

The proposed access algorithm limits the time spent by packets at the node and, as a result, eliminates the non-stationary behavior of the network. However, when using it, information packets may be lost.

The network simulation results for the considered access method are shown in Figures 9 and 10.

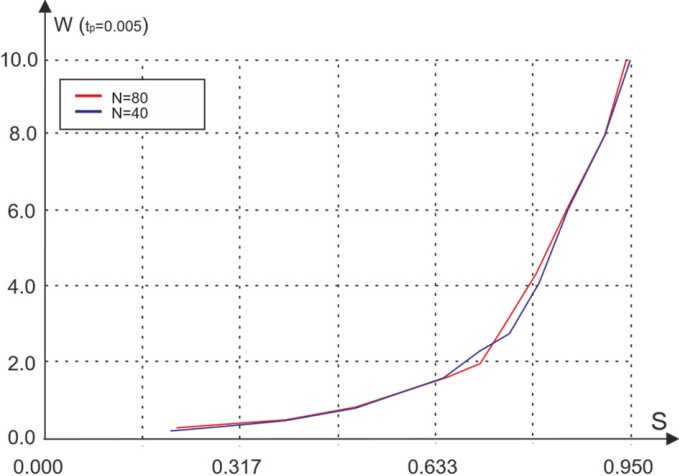

**Figure 9.** Dependence of the average expectation on the amount of traffic.

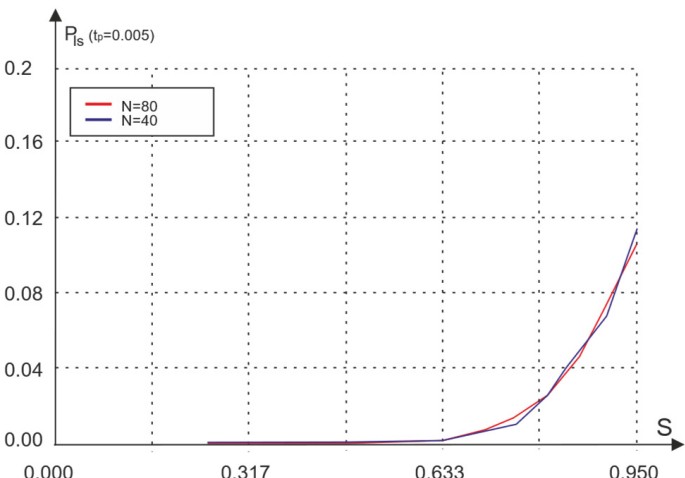

**Figure 10.** Dependence of the packet loss probability on the amount of traffic.

The analysis of the obtained results shows that the average waiting time for access with the last collision acknowledgment is comparable to the waiting time for basic access. However, in contrast to the basic access algorithm, the characteristics of access with the acknowledgment of the last distortion are practically independent of the size of the receipt. The probability of packet loss during access with last collision acknowledgment is significantly lower than the probability of loss without acknowledgment (Figure 3), which is especially noticeable for low and medium traffic.

Figure 11 shows the dependence of the coefficient of variation in waiting time on traffic. As follows from this figure, the change in the coefficient of variation is nonmonotonic. The growth section is limited by the network capacity for a given propagation time $t_p$, and the probability of packet loss is also small in this section. The section of decrease in the coefficient of variation corresponds to the range of traffic where there is a sharp increase in the probability of packet loss.

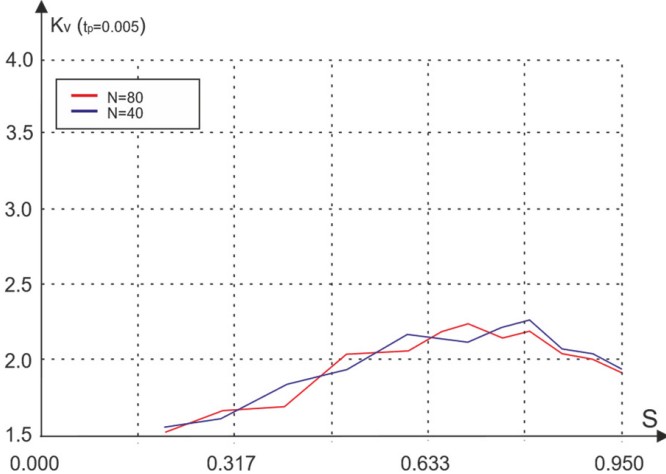

**Figure 11.** Dependence of the coefficient of variation in waiting time on traffic for different numbers of nodes.

Last distortion acknowledgement access provides a high network capacity and a relatively low level of packet loss, which makes it interesting for practical applications. However, its efficiency can be further improved by controlling DM distortion.

*4.4. Analysis of the Characteristics of the Access Method with Acknowledgment of Distortions of DM*

In this case, DM is acknowledged by some node (for example, a randomly selected one). The implementation of the access algorithm is as follows. When a collision is detected, the gateway sends DM. If the message is transmitted without distortion, then it is accepted by all nodes. One of the sensors transmits DM receipt, the other sensors wait for the end of its transmission. After that, the normal functioning of the network continues: primary and previously distorted packets are transmitted. If DM is malformed, then the gateway does not receive an acknowledgment and retransmits the message according to the general random access procedure.

The proposed access algorithm, on the one hand, limits the retransmission waiting time and, on the other hand, eliminates packet loss. The simulation results of the considered access mechanism are shown in Figures 12 and 13. It should be noted that the characteristics of this access method, which can be used without restrictions on the size of the packet or receipt, are practically the same as the characteristics of the basic CSMA/CA method.

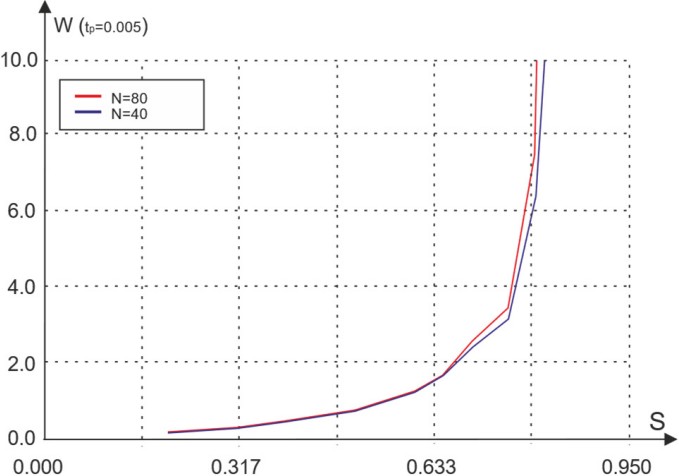

**Figure 12.** Dependence of the average waiting time on traffic for different numbers of nodes.

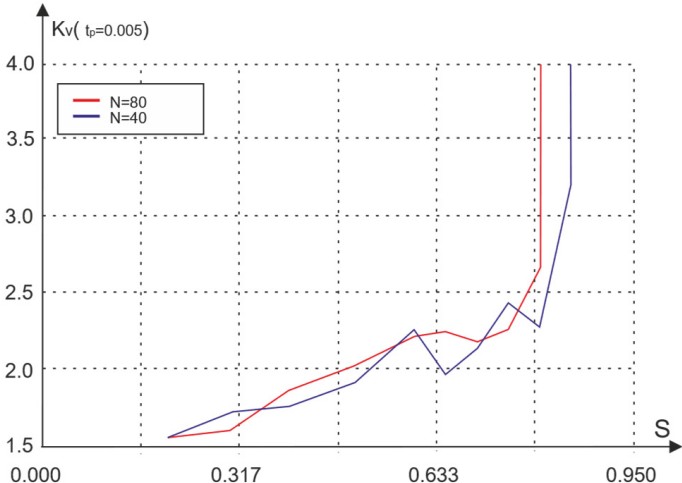

**Figure 13.** Dependence of the coefficient of variation in waiting time on traffic for different numbers of nodes.

## 5. Discussion

Based on the analysis of the simulation results, it can be concluded that the efficiency of the transmission of short information packets in IoT networks can be improved in several ways, the choice of which is determined by the amount of traffic and network load. If the

load is small, then it may be interesting to use the simplest access protocols that allow for packet loss. Due to low traffic, such losses may be acceptable. If the losses are unacceptable and the network load is high, it is necessary to use modified random access protocols based on the centralized generation of special DM. The complication of access protocols in this case is offset by an increase in network efficiency.

The above results refer to only one signal propagation time, $t_p = 5 \times 10^{-3}$. A change in $t_p$ quantitatively affects only the network capacity, while, as is shown by the simulation results, the coefficient of variation and the probability of packet loss are practically independent of the propagation time.

Another important remark is the following: the carrier listening access protocol has an additional control parameter (the interval for waiting for a packet to be transmitted after the channel is released (interframe gap)). The analysis shows that the behavior of the network is sensitive to the value of this parameter, and for efficient operation, it is necessary to use optimal control.

Figure 14 shows dependence of the optimal waiting interval $Z_{OPT_i}$ on the amount of input traffic and the number of nodes for access with acknowledgment of DM distortions.

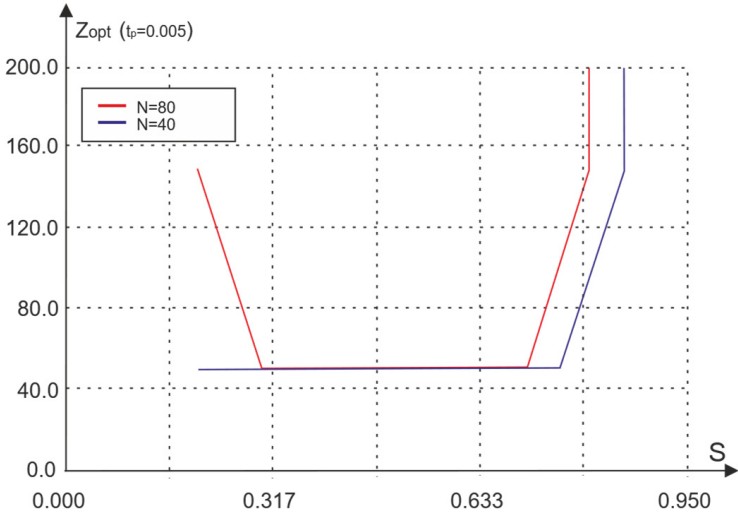

**Figure 14.** Dependence of the optimal interframe gap on the amount of input traffic and the number of nodes.

As follows from the presented results, the task of managing the optimization of access parameters is not trivial, since it is necessary to take into account the dependence of the $Z$ interval on the amount of current traffic, the number of nodes and the signal propagation time. In this regard, it should be noted that the results presented above are obtained with optimal control of the waiting interval.

## 6. Conclusions

The article considers options for organizing multiple random access under the condition of transmitting short information packets, the size of which is comparable to the size of receipts. It is shown that traditional access protocols under these conditions can lead to a significant decrease in network capacity.

On the basis of simulation modeling, the characteristics were determined and a comparative analysis of the basic CSMA/CA access algorithm, access algorithm without acknowledgement and new access algorithms was carried out. In the new access algorithms, in the event of collision detection, a dedicated node generates and broadcasts a special DM in the broadcast mode, on the basis of which the network nodes repeatedly send garbled packets.

It is shown that for the proposed access algorithms, the size of the information packet does not significantly affect the network capacity. At the same time, the lack of a complete

acknowledgment either leads to an unstable network operation or does not completely prevent the loss of information packets.

To eliminate this shortcoming, an access algorithm is proposed in which DM is additionally acknowledged. In this case, it is possible to ensure high network capacity, eliminate losses and achieve independence of network characteristics from the size of receipts and information packets.

The proposed modification of the basic access protocol for IoT networks makes it possible to increase the network capacity (the number of data sources served) depending on the initial traffic by 1.5–2 times or reduce the waiting time for packet transmission by 1.5–2 times (while maintaining the capacity of the base case). The conducted studies allow concluding that the modification of traditional radio network access mechanisms to take into account the peculiarities of IoT networks can significantly increase their efficiency.

Therefore, further work in this direction is also important. New directions include the following:

1. Study of the characteristics of heterogeneous networks that combine mobile networks and IoT networks. Interest in such research is due to the fact that in heterogeneous networks, the size of information packets can be either comparable to the size of receipts or much larger.

2. Study of the characteristics of networks with non-stationary flows in time, which are characterized by the presence of time periods with significantly different frequency of packet generation.

3. Study of IoT network management algorithms that reduce the risks of their transition to an unstable state.

4. Accounting for the impact of external noise, primarily industrial interference, on the efficiency of IoT network.

**Author Contributions:** Conceptualization, V.V.B., V.E.K. and V.A.S.; methodology, V.V.B., V.E.K. and V.A.S.; software, V.V.B., V.E.K. and V.A.S.; validation, V.V.B., V.E.K. and V.A.S.; formal analysis, V.V.B., V.E.K. and V.A.S.; investigation, V.V.B., V.E.K. and V.A.S.; resources, V.V.B., V.E.K. and V.A.S.; data curation, V.V.B., V.E.K. and V.A.S.; writing—original draft preparation, V.V.B., V.E.K. and V.A.S.; writing—review and editing, V.V.B., V.E.K. and V.A.S.; visualization, V.V.B., V.E.K. and V.A.S.; supervision, V.V.B., V.E.K. and V.A.S.; project administration, V.V.B., V.E.K. and V.A.S.; funding acquisition, V.V.B., V.E.K. and V.A.S. All authors have read and agreed to the published version of the manuscript.

**Funding:** This research was funded by Russian Science Foundation, grant number 23-69-10084, https://rscf.ru/project/23-69-10084/, accessed on 15 May 2023.

**Data Availability Statement:** Not applicable.

**Conflicts of Interest:** The authors declare no conflict of interest.

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
