# Peer review of "Analysis of the Efficiency of Various Receipting Multiple Access Methods with Acknowledgement in IoT Networks"

_inventions, doi:10.3390/inventions8040105_

Round 1

Reviewer 1 Report

This article investigates the implementation methods of CSMA/CA access methods with various acknowledgement options and without acknowledgement to reduce packet transmission overhead, addressing the issue of receipt transmission requiring a large amount of channel bandwidth and reducing network efficiency when the sizes of information packets are small and comparable to the sizes of receipts. Overall, the article provides a more comprehensive description.
However, there are still some issues worth recommending:
1. In the abstract, it is described that the proposed implementation methods of CSMA/CA access method with various acknowledgement options and without acknowledgement can significantly reduce packet transmission overhead. Which method should be clarified? And provide clear numerical results for the efficiency indicator.
2. The focus of the article is to study the situation where the size of the sizes of information packets are small and comparable to the sizes of receipts, and is not insensitive to data size. Please provide an explanation, especially in the introduction, which should be clear.
3. Regarding the implementation methods of CSMA/CA access method with various acknowledgement options and without acknowledgement mentioned in the article, including basic CSMA/CA access algorithms, unconfirmed access algorithms, and new access algorithms, a comparison with other studies should be provided in the conclusion to indicate the improvement in efficiency.
4. It is recommended to express the curves in the text more clearly, such as coordinate units, curve clarity, etc.

The overall English quality of the article is not a problem, so it can be carefully checked and modified appropriately.

Author Response

1. In the abstract, it is described that the proposed implementation methods of CSMA/CA access method with various acknowledgement options and without acknowledgement can significantly reduce packet transmission overhead. Which method should be clarified? And provide clear numerical results for the efficiency indicator.

Answer: We made changes in the sections "Abstract" and "Conclusions".

2. The focus of the article is to study the situation where the size of the sizes of information packets are small and comparable to the sizes of receipts, and is not insensitive to data size. Please provide an explanation, especially in the introduction, which should be clear.

Answer: We made changes in the section "Introduction": it is noted that the task of transmitting short packets is relevant not only for IoT networks based on the IEEE 802.15.4 standard, but also for a number of other applications. The purpose of the study and methods for achieving it are formulated.

3. Regarding the implementation methods of CSMA/CA access method with various acknowledgement options and without acknowledgement mentioned in the article, including basic CSMA/CA access algorithms, unconfirmed access algorithms, and new access algorithms, a comparison with other studies should be provided in the conclusion to indicate the improvement in efficiency.

Answer: We made changes in the section “Conclusions” indicating the numerical gain in network capacity and latency compared to the basic CSMA/CA access method.

4. It is recommended to express the curves in the text more clearly, such as coordinate units, curve clarity, etc.

Answer: The figures were corrected according the reviewer’s comment.

Reviewer 2 Report

A set of Receipting Multiple Access methods with  and without acknowledgement are anlyzed.

In the introductory section, a concise description of the performance objectives of the research is missing. The authors must briefly highlight what these objectives are and how they intend to pursue them.

The CSMA/CA algorithm needs to be presented in a more structured way; for easier reading it is advisable to structure CMSA/CA in pseudocode mode.

The description of the model given in paragraph 3.1 must also be shown in a more structured way. It would be appropriate for the authors to show the architectural scheme of the model in a figure, including its composition in the 3 blocks.

What are the future research developments? The authors should add these considerations in the concluding section.

An extensive editing of English is required.

Author Response

1) In the introductory section, a concise description of the performance objectives of the research is missing. The authors must briefly highlight what these objectives are and how they intend to pursue them.

Answer: We made changes in the section "Introduction": the purpose of the study and methods for achieving it are clearly formulated.

2) The CSMA/CA algorithm needs to be presented in a more structured way; for easier reading it is advisable to structure CMSA/CA in pseudocode mode.

Answer: The authors did not set the task of presenting CSMA/CA access method in a structured form, since this issue, as applied to wireless IoT networks, is quite fully covered in a number of works. But taking into account the comment of the reviewer, we made an appropriate explanation in the text of the article and added two articles, which details the algorithm of CSMA/CA in IoT networks. Accordingly, changes have been made to the list of references:

Chen X., Shu Z., Wang K., Xu F., Cao Y. Proportional Fairness in Wireless Powered CSMA/CA Based IoT Networks// 2018 IEEE Global Communications Conference (GLOBECOM), Abu Dhabi, United Arab Emirates, 2018, pp. 1-7, doi: 10.1109/GLOCOM.2018.8648072.

Shafiq M., Choi J-G. MSMA/CA: Multiple Access Control Protocol for Cognitive Radio-Based IoT Networks// Journal of Internet Technology, vol. 20, no. 1, pp. 301-313, Jan. 2019.

3) The description of the model given in paragraph 3.1 must also be shown in a more structured way. It would be appropriate for the authors to show the architectural scheme of the model in a figure, including its composition in the 3 blocks.

Answer: Taking into account the fact that the simulation model is described in sufficient detail in one of the authors' works (to which there is a link in the text of the article), the authors do not consider it appropriate to pay too much attention to it in this article. Section 2.2 lists the 3 main functional blocks and gives a brief description of them.

But taking into account the reviewer's comment, the authors made adjustments to the text of the article and added the necessary explanations to section 2.2

4) What are the future research developments? The authors should add these considerations in the concluding section.

Answer: We made changes in the section “Conclusions”, in particular, directions for further research have been formulated.

Round 2

Reviewer 2 Report

The authors have revised their manuscript in detail taking into account all my suggestions. I consider this paper publishable in the current form.